# Thrombopoietin mutation in congenital amegakaryocytic thrombocytopenia treatable with romiplostim

Alessandro Pecci[1], Iman Ragab[2], Valeria Bozzi[1], Daniela De Rocco[3], Serena Barozzi[1], Tania Giangregorio[4], Heba Ali[2], Federica Melazzini[1], Mohamed Sallam[5], Caterina Alfano[6,7], Annalisa Pastore[8], Carlo L Balduini[1] & Anna Savoia[3,4,*] iD

## Abstract

Congenital amegakaryocytic thrombocytopenia (CAMT) is an inherited disorder characterized at birth by thrombocytopenia with reduced megakaryocytes, which evolves into generalized bone marrow aplasia during childhood. Although CAMT is genetically heterogeneous, mutations of *MPL*, the gene encoding for the receptor of thrombopoietin (THPO), are the only known disease-causing alterations. We identified a family with three children affected with CAMT caused by a homozygous mutation (p.R119C) of the *THPO* gene. Functional studies showed that p.R119C affects not only ability of the cytokine to stimulate MPL but also its release, which is consistent with the relatively low serum THPO levels measured in patients. In all the three affected children, treatment with the THPO-mimetic romiplostim induced trilineage hematological responses, remission of bleeding and infections, and transfusion independence, which were maintained after up to 6.5 years of observation. Recognizing patients with *THPO* mutations among those with juvenile bone marrow failure is essential to provide them with appropriate substitutive therapy and prevent the use of invasive and unnecessary treatments, such as hematopoietic stem cell transplantation or immunosuppression.

**Keywords** congenital amegakaryocytic thrombocytopenia; MPL; mutation; romiplostim; thrombopoietin
**Subject Categories** Development & Differentiation; Genetics, Gene Therapy & Genetic Disease; Pharmacology & Drug Discovery

See also: **AR Kim & VG Sankaran** (January 2018)

## Introduction

Congenital amegakaryocytic thrombocytopenia (CAMT, MIM 604498) is an inherited disorder characterized at birth by severe thrombocytopenia with reduced/absent bone marrow megakaryocytes, and subsequently evolution, usually within the first decade of life, into pancytopenia due to trilineage bone marrow aplasia (Ballmaier *et al*, 2003; Ballmaier & Germeshausen, 2011). CAMT is fatal unless children are treated with allogeneic hematopoietic stem cell transplantation (HSCT).

The majority of CAMT cases are caused by biallelic mutations in the gene (*MPL*) encoding for the myeloproliferative leukemia virus oncogene (MPL) receptor, which abolish or severely impair the interaction of the receptor with its ligand thrombopoietin (THPO) (Ihara *et al*, 1999; Ballmaier *et al*, 2001; Savoia *et al*, 2007; Ballmaier & Germeshausen, 2009). The clinical course of CAMT reflects the non-redundant role of the THPO/MPL pathway in megakaryocyte and platelet production since birth and in maintenance of the multipotent stem cell compartment in the post-natal hematopoiesis (Ballmaier & Germeshausen, 2011; Hirata *et al*, 2013). Of note, a proportion of patients with the clinical picture of CAMT does not carry mutations in *MPL*, suggesting that the disease is genetically heterogeneous. It has been hypothesized that the patients with normal MPL have alterations in genes playing a role upstream or downstream of the receptor (Ballmaier & Germeshausen, 2011; Geddis, 2011). Considering that mice lacking Thpo have a similar phenotype as those lacking Mpl (Carver-Moore *et al*, 1996), *THPO* is a strong candidate.

A homozygous missense mutation in *THPO* (p.R38C) was previously reported in a family with a recessive form of aplastic anemia (Dasouki *et al*, 2013). Very recently, two other homozygous *THPO* mutations (p.R99W and p.R157*) have been associated with

1  Department of Internal Medicine, IRCCS Policlinico San Matteo Foundation and University of Pavia, Pavia, Italy
2  Hematology-Oncology Unit, Pediatric Hospital, Ain Shams University, Cairo, Egypt
3  Institute for Maternal and Child Health – IRCCS Burlo Garofolo, Trieste, Italy
4  Department of Medical Sciences, University of Trieste, Trieste, Italy
5  Department of Clinical Pathology, Ain Shams University, Cairo, Egypt
6  Maurice Wohl Clinical Neuroscience Institute, King's College, London, UK
7  Fondazione Ri.MED, Palermo, Italy
8  Department of Molecular Medicine, University of Pavia, Pavia, Italy
   *Corresponding author. Tel: +39 040 3785527; Fax: +39 040 3785540; E-mail: anna.savoia@burlo.trieste.it

inherited bone marrow failure syndrome in three unrelated pedigrees. Two affected individuals from these families were successfully treated by administration of the MPL agonist romiplostim (Seo *et al*, 2017).

Here, we report a family where the clinical picture of CAMT is caused by a novel homozygous mutation in *THPO*. We show that the mutation significantly impairs secretion of the cytokine. Moreover, activity of the mutant THPO was reduced by about 50%, consistent with defective stimulation of the signaling pathways downstream of the MPL receptor. The three affected children were all treated with romiplostim, resulting in sustained trilineage response, which was maintained after an up to 6.5-year follow-up.

# Results

## Clinical features of the family and response to romiplostim of the affected children

The propositus is a male born to first cousin parents of Egyptian origin who presented in July 2009, at the age of 3.5 years, with ecchymotic patches. His physical examination showed pre-auricular skin tags and left eye divergent squint. No other physical abnormalities were found. Blood cell count revealed very severe thrombocytopenia, hyporegenerative anemia (reticulocyte count $22.8 \times 10^9$/l), and severe neutropenia (Table 1). Microscopy evaluation of peripheral blood films did not identify any cytomorphological abnormalities, and platelet size was normal (Noris *et al*, 2014). Bone marrow examination showed hypocellularity with very rare immature megakaryocytes, and conventional marrow cytogenetics showed normal karyotype. After exclusion of Fanconi anemia (no spontaneous and diepoxybutane-induced chromosome breakage) and paroxysmal nocturnal hemoglobinuria (normal CD55/CD59 expression in granulocytes and erythrocytes), a diagnosis of idiopathic aplastic anemia was made.

Shortly after, the propositus started to suffer from recurrent major hemorrhages (epistaxis, gum, and rectal bleeding) and recurrent severe infectious episodes (febrile neutropenia or otitis media occurring every 1–2 months), which required frequent hospital admissions and multiple platelet and red blood cell transfusions.

Due to the severity of the clinical picture, HSCT was considered and proband's relatives were investigated. Blood cell counts were normal in the parents, while severe isolated thrombocytopenia was found in the 7-month-old brother (Table 1). In this subject, peripheral blood cell morphology was normal and bone marrow examination showed a remarkably decreased number of megakaryocytes without other significant abnormalities.

Given the lack of a family donor for HSCT, the propositus was treated with cyclosporine A for 4 months, without any response, and then with oxymetholone, again without any improvement. Due to the severe bleeding symptoms and the need for very frequent platelet transfusions, as well as to the lack of other available therapeutic options, in June 2010, at the age of 4.4 years, the propositus was started on empirical treatment with romiplostim. Romiplostim was given at the dose of 1 μg/kg/week for 3 months with clear and rapid trilineage improvement of pancytopenia. Afterward, because of difficulties in accessing to healthcare facilities, the patient received the dose of 4 μg/kg once a month. Figure 1 shows hemoglobin concentration, leukocyte count, and platelet count before and during romiplostim administration. After starting romiplostim, all of these parameters were stably above the values observed before treatment. The differential count was performed only on some visits: The mean neutrophil count was $0.7 \times 10^9$/l (range 0.3–1.1) in the 12 measurements before romiplostim and $1.7 \times 10^9$/l (range 1.1–2.9) in the eight available measurements during treatment ($P < 0.005$). Most importantly, no further spontaneous bleeding events and infectious episodes were recorded and no further transfusion needed during romiplostim administration. The patient is currently on treatment, and the hematological and clinical response is maintained after a follow-up of 6.5 years.

At the age of 1.6 years, the proband's brother started to present with recurrent spontaneous bleeding requiring platelet transfusions. In view of the good response obtained in the propositus, in September 2010, he was started on romiplostim (4 μg/kg once a month). He achieved good platelet response (platelet count between 30 and $100 \times 10^9$/l) and remission of bleedings with stable independence from transfusions that was maintained after a 6.3-year follow-up. Recently, the proband's parents gave birth to a third, apparently normal child. At the age of 2 months, she developed a hematoma at the site of vaccination. Blood cell counts revealed anemia and severe thrombocytopenia with normal blood cell morphology. She

**Table 1. Clinical features and blood counts of the proband and family members at the first examination.**

|  | Father | Mother | Proband | 1st sibling | 2nd sibling |
|---|---|---|---|---|---|
| Age (months)/Gender |  |  | 42/M | 7/M | 2/F |
| Clinical presentation | None | None | Spontaneous bleeding, infections | None | Bleeding at vaccination |
| Hemoglobin (g/dl) | 13.6 | 12.3 | 7.8 | 12.5 | 8.2 |
| Red blood cells ($\times 10^{12}$/l) | 4.27 | 3.83 | 2.28 | 4.27 | 2.71 |
| Mean corpuscular volume (fl) | 91.4 | 91.6 | 100 | 83.4 | 84.9 |
| White blood cells ($\times 10^9$/l) | 5 | 9.6 | 5.4 | 11.7 | 9.5 |
| Neutrophils ($\times 10^9$/l) | 2 | 6.6 | 0.3 | 3.34 | 2.7 |
| Platelets ($\times 10^9$/l) | 197 | 193 | 3 | 26 | 27 |
| Platelet diameter, mean/2.5th–97.5th (μm)[a] | 3.3/2.3–5.2 | 2.7/1.6–4.8 | 3.1/1.8–4.9 | 2.6/1.6–3.9 | 2.7/1.9–4.2 |

[a]Normal values: 2.6/1.6–3.9 μm (Noris *et al*, 2014).

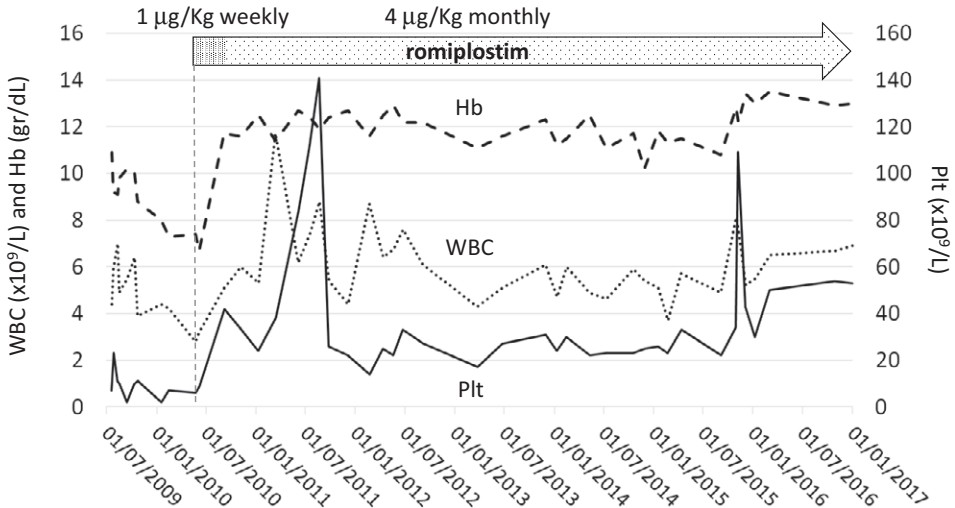

**Figure 1.  Time course of hemoglobin level (Hb), white blood cell count (WBC), and platelet count (Plt) in the propositus before and during romiplostim administration.**

Romiplostim was started at the dosage of 1 μg/kg/week, and this schedule was maintained for 3 months. Then, a dose of 4 μg/kg was given in a monthly administration. The platelet count had wide oscillations that derived, at least partially, from the variability of the time elapsed between the drug administrations and the measurements of blood count.

was started on romiplostim in October 2015, with good platelet and hemoglobin response. She is presently receiving romiplostim 4 μg/kg monthly with platelet counts ranging between 30 and 60 × 10⁹/l and hemoglobin concentration between 9.9 and 12.3 g/dl. She has no spontaneous bleeding.

Due to logistic difficulties, platelet count was not regularly measured at fixed times during romiplostim treatment. For this reason, wide variations in platelet counts were observed in all the treated children during the monthly administration (Fig 1). In Table 2, we report the changes in blood counts observed 1 week after the monthly administration of romiplostim (4 μg/kg), showing that the short-term effect of the treatment was a consistent significant increase in platelet counts. No side effects attributable to romiplostim were observed in any of the three children.

### Identification of the p.R119C THPO in the pedigree

Based on the clinical pictures of the three patients suggesting a defect of the THPO/MPL pathway, we screened for mutation both the *MPL* and *THPO* genes in the proband. While no variants were found in *MPL*, we identified the homozygous c.355C>T variant in *THPO* (p.R119C). Consistent with recessive inheritance of the phenotype, the proband's siblings were homozygous and the

parents were heterozygous for the variant (Fig EV1). The p.R119C is not reported in the SNP (ncbi.nlm.nih.gov/SNP), 1000 Genome (1000genomes.org), and ExAC (exac.broadinstitute.org) databases. The potential effect of the amino acid substitution on protein function was evaluated using different pathogenicity prediction programs, such as CADD (score 32), Mutation Taster (disease-causing mutation), Mutation Assessor (neutral), and SIFT (not tolerated), suggesting that p.R119C is likely to be pathogenic.

### The p.R119C is predicted to affect THPO interaction with MPL

Structural bioinformatic analysis showed that residue R119 (R98 in the mature protein after removal of the leader peptide) is within the N-terminal region that is involved in the binding of THPO to its receptor MPL. In particular, the binding domain of THPO is a 4-helices ribbon bundle domain and R119 resides in the middle of helix C (Fig EV2). This residue is highly conserved not only throughout species but also in the receptor binding domain of the homologous human erythropoietin (EPO), which shares 23% sequence homology with THPO (Fig EV2).

We analyzed the structure of the receptor binding domain of THPO in complex with a neutralizing antibody fragment (PDB 1V7M) (Feese *et al*, 2004). In this structure, although R119 is not

**Table 2.  Representative changes in blood counts 1 week after monthly administration of romiplostim 4 μg/kg.**

|  | Proband | | 1st sibling | | 2nd sibling | |
|---|---|---|---|---|---|---|
|  | **Before** | **After** | **Before** | **After** | **Before** | **After** |
| Hemoglobin (g/dl) | 12.3 | 13.4 | 12.6 | 12.7 | 8.8 | 9.7 |
| White blood cells (×10⁹/l) | 5.2 | 6.6 | 8.4 | 8.7 | 11.3 | 10.5 |
| Neutrophils (×10⁹/l) | 1.5 | 1.6 | 2.4 | 2.9 | 1.7 | 2.6 |
| Platelets (×10⁹/l) | 43 | 94 | 28 | 86 | 20 | 65 |

    

involved in intermolecular interactions that could be essential for the correct folding of the domain, it forms a salt bridge with the side chain of D31 of the antibody fragment (Fig EV2). A similar interaction is also observed in the structure of EPO in a complex with its receptor. In this case, the analogous arginine (R103) is involved in several polar interactions that make this residue a key player for binding to its receptor (Fig EV2) (PDB 1EER) (Syed *et al*, 1998). Therefore, it is reasonable to hypothesize that the p.R119C mutation does not affect the folding of THPO but the interaction with MPL.

### The p.R119C affects both secretion of THPO and THPO/MPL signaling

We investigated the consequences of p.R119C on the THPO function using the UT7-TPO human megakaryoblastic leukemia cells, an engineered line stably expressing MPL that completely depends on THPO for growth and survival (Komatsu *et al*, 1996; Komatsu, 2004). We first transfected the wild-type or p.R119C THPO in HEK293T cells to produce THPO-conditioned culture supernatants. We also tested the p.R38C variant that was previously identified in a family with recessive aplastic anemia to compare the effects of the two substitutions (Dasouki *et al*, 2013). Immunoblotting showed that the three exogenous THPO forms were efficiently expressed in HEK293T cells, with the mutant proteins expressed even at slightly higher levels than the wild type (Fig 2A–D).

We then probed the ability of the different THPO-conditioned HEK293T culture supernatants in sustaining the growth of the UT7-TPO cells. After incubation with medium containing equal amounts (0.5 or 1.0 μl) of wild-type or mutant supernatants, the UT7-TPO proliferation induced by both mutant supernatants was strongly reduced compared to that promoted by the wild type (Fig 3). Therefore, we measured the THPO concentrations in the HEK293T culture supernatants and found that the THPO concentration in the conditioned media containing both mutants was much lower than that in the wild-type sample (about 23-fold lower for both the p.R119C and the p.R38C) (Table 3). Since the THPO expression levels in the HEK293T cellular lysates were similar between wild type and mutants (or even higher for the mutant forms) (Fig 2A–D), we hypothesized that the two variants interfere with the secretion of THPO from the intracellular to the extracellular compartment.

To investigate whether the p.R119C or p.R38C substitutions could instead affect stability of THPO protein, we studied stability of wild-type and mutant forms using the cycloheximide chase assay (Tétreault *et al*, 2016; Zhang *et al*, 2016). After blocking *de novo* protein synthesis in HEK293T cells by treatment with cycloheximide, the kinetics of degradation of the wild-type and mutant THPOs were assessed over the subsequent 48 h. The assay showed that the mutant proteins have similar stability to that of the wild-type form (Fig 4A and B).

To investigate the ability of the p.R119C mutant in stimulating MPL, we then performed the proliferation assay by incubating UT7-TPO cells with the same concentrations of wild-type or mutant THPO. We found that the p.R119C THPO induced a significantly lower UT7-TPO proliferation compared to the wild type at each of the three different THPO concentrations tested (60, 120, or 240 pg/ml) (Fig 5). The mean activity of the p.R119C was $52\% \pm 7$ ($P < 0.05$), $53\% \pm 9$ ($P < 0.05$), and $45\% \pm 4$ ($P < 0.005$) of the wild-type THPO, respectively. Similar results were obtained with the p.R38C mutant (Fig 5).

Finally, to support the results obtained with the proliferation assay, we stimulated UT7-TPO cells with wild-type or mutant HEK293T supernatants and investigated the phosphorylation status of key signaling kinases downstream of the MPL receptor, such as STAT5, ERK1/2, and AKT (Hitchcock & Kaushansky, 2014). The analysis confirmed that both p.R119C and p.R38C THPO induce defective activation of all the investigated intracellular signaling kinases (Fig 6A and B).

### Serum THPO levels are not increased in patients

Thrombopoietin levels measured in the sera of the two older siblings collected before romiplostim administration were within the normal range. In particular, THPO concentration was 10.9 pg/ml in the proband and 14.1 pg/ml in his brother. The serum THPO levels obtained in 50 consecutive healthy subjects ranged from 6.9 to 54.4 pg/ml (mean: 14.6) (Noris *et al*, 2011). Since the serum THPO level is negatively regulated by megakaryocytes/platelet mass, it is always markedly increased in all forms of bone marrow aplasia-hypoplasia, including the CAMT form caused by *MPL* mutations (Dame, 2001; Ballmaier & Germeshausen, 2009; Olnes *et al*, 2012), further supporting the hypothesis that production of THPO is defective in our patients.

---

**Figure 2.  Transfection of constructs harboring the wild-type or mutant *THPO* cDNAs induced efficient expression of THPO proteins in HEK293T cells.**

A   Representative images of immunoblotting of HEK293T cell lysates prepared 24 and 48 h after transfection with wild-type (WT) or mutant (p.R119C and p.R38C) *THPO*-expressing or empty vectors and then pooled into a ratio of 1:1 (Dasouki *et al*, 2013). THPO cDNAs were tagged with the FLAG epitope. β-Actin was used as loading control.

B   Densitometric analysis of the bands obtained by immunoblotting of HEK293T cell lysates prepared 24 and 48 h after transfection and then pooled into a ratio of 1:1. Immunoblotting was performed on cells collected after $n = 3$ independent transfection experiments. THPO levels are expressed as FLAG/β-actin ratio and THPO/β-actin ratio.

C   Representative images of immunoblotting of HEK293T cell lysates prepared 24 and 48 h after transfection with wild-type or mutant *THPO*-expressing or empty vectors and then analyzed separately (24 and 48 h).

D   Densitometric analysis of the bands obtained by immunoblotting of HEK293T cell lysates prepared 24 and 48 h after transfection and then analyzed separately. Immunoblotting was performed on cells collected after $n = 3$ independent transfection experiments. Samples derive from the same experiment, and blots were processed in parallel.

Data information: In (B, D), data are presented as means ± SD. The asterisk (*) indicates a *P*-value < 0.05 with respect to WT (two-tailed Student's *t*-test). The exact *P*-values are reported in Appendix Table S1.
Source data are available online for this figure.

**A**

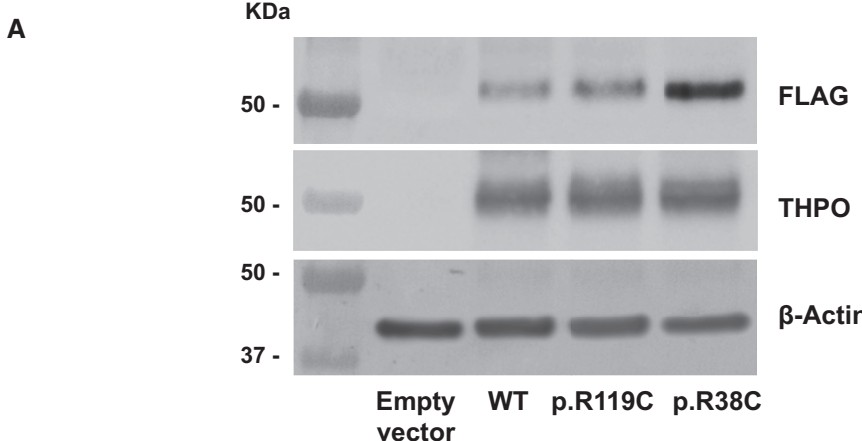

**B**

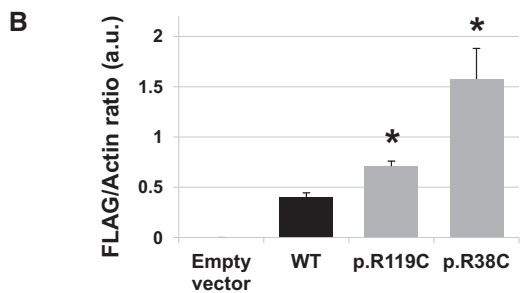
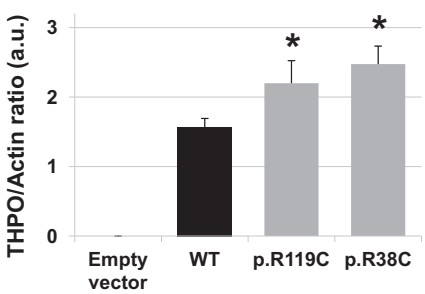

**C**

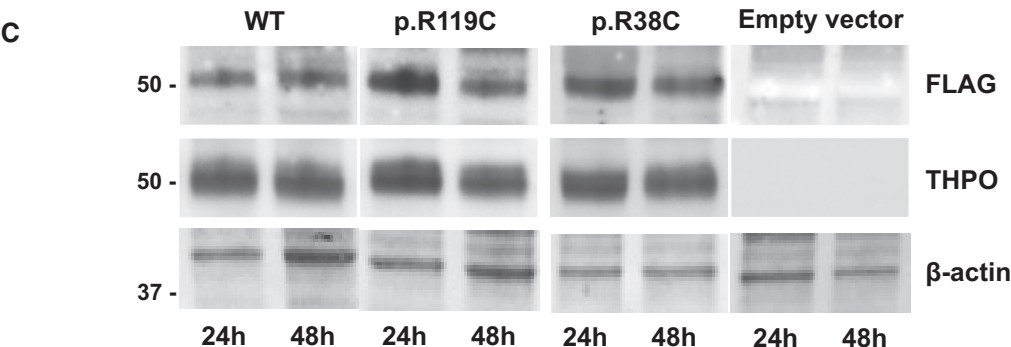

**D**

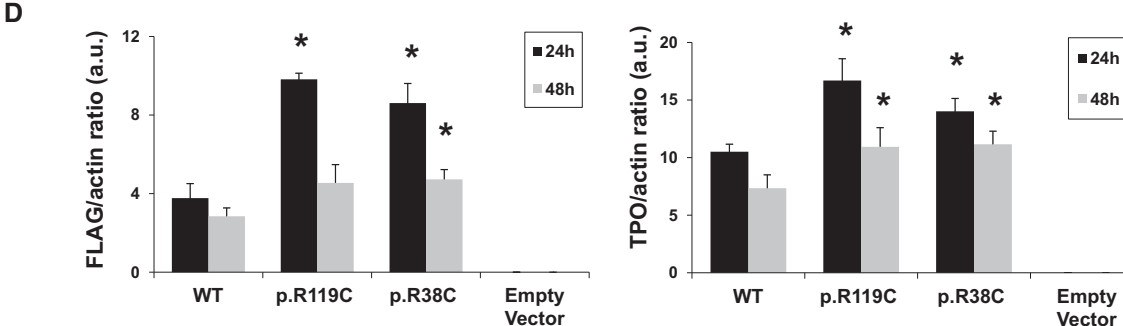

Figure 2.

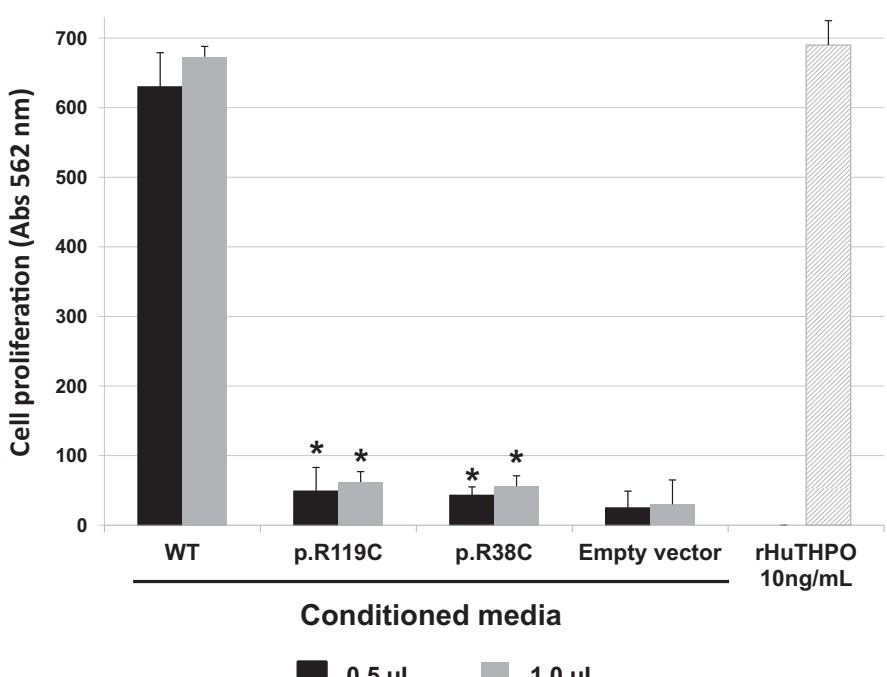

**Figure 3.  The mutant THPO-conditioned culture supernatants are markedly deficient in sustaining proliferation of UT7-TPO cells.**

Cell proliferation measured in $n = 3$ independent MTT assays and expressed as absorbance units of UT7-TPO cells incubated with medium containing 0.5 or 1.0 µl of THPO-conditioned supernatants from HEK293T cells transfected with wild-type (WT) or mutant (p.R119C and p.R38C) *THPO*-expressing or empty vectors. Recombinant human THPO (rHuTHPO) at the concentration of 10 ng/ml was also used as control. Data are presented as means $\pm$ SD. The asterisk (*) indicates a *P*-value < 0.01 with respect to WT (two-tailed Student's *t*-test). The exact *P*-values are reported in Appendix Table S2.

Source data are available online for this figure.

**Table 3.  THPO concentrations measured in the conditioned supernatants from HEK293T cells transfected with wild-type or mutant *THPO*-expressing or empty vectors.**

|  | THPO (ng/ml) (mean $\pm$ SD) |
|---|---|
| Wild-type THPO | 239.9 $\pm$ 11.4 |
| p.R119C THPO | 10.6 $\pm$ 2.4[a] |
| p.R38C THPO | 10.7 $\pm$ 3.9[a] |
| Empty vector | 0 |

Supernatants were collected 24 and 48 h after transfection and then pooled in a ratio of 1:1 (Dasouki *et al*, 2013).
[a]*P* < 0.05 with respect to wild-type THPO.

## Discussion

In humans, the THPO/MPL pathway is essential not only for platelet production since birth but also for preservation of the integrity of the multipotent stem cell compartment in the post-natal hemopoiesis (Ballmaier & Germeshausen, 2011; Hirata *et al*, 2013; Hitchcock & Kaushansky, 2014). The phenotype of CAMT is the most convincing proof of these non-redundant and non-synchronous roles of this pathway. In fact, patients with CAMT not only have congenital hypo-megakaryocytic thrombocytopenia but also develop during childhood further cytopenias until progression to generalized bone marrow aplasia. At present, loss-of-function mutations in *MPL* are the only known cause of CAMT, although some observations

indicate that the disorder is genetically heterogeneous (Ballmaier & Germeshausen, 2011; Geddis, 2011).

In the Egyptian pedigree reported in this paper, we show that a homozygous loss-of-function variant in *THPO* (p.R119C) induces a phenotype similar to that caused by *MPL* mutations. Indeed, the oldest of the three affected siblings presented with trilineage bone marrow hypoplasia, while the younger ones had isolated thrombocytopenia or thrombocytopenia with anemia. Thus, since thrombocytopenia was the only feature found in all the patients, the disorder could, in the first instance, be regarded as an isolated thrombocytopenia or with thrombocytopenia variably associated with anemia and/or neutropenia. However, the trilineage aplasia observed in the oldest sibling suggests that the hematological picture worsens with age. As a matter of fact, the clinical presentation of these patients appears indistinguishable from that of subjects with CAMT due to MPL mutations.

The first homozygous loss-of-function variant of *THPO* (p.R38C) was reported in a Micronesian family presenting with a recessive form of aplastic anemia (Dasouki *et al*, 2013). Bone marrow aplasia was recognized at the age of 16 and 28 years in the two investigated patients, and family history revealed that two siblings had died for aplastic anemia in their adolescence. The paper did not report whether in these subjects, bone marrow aplasia was preceded by a history of congenital thrombocytopenia. Therefore, we cannot exclude that also the Micronesian patients had a clinical course similar to that of CAMT, even if globally less severe, given that marrow aplasia was diagnosed during their adolescence or later.

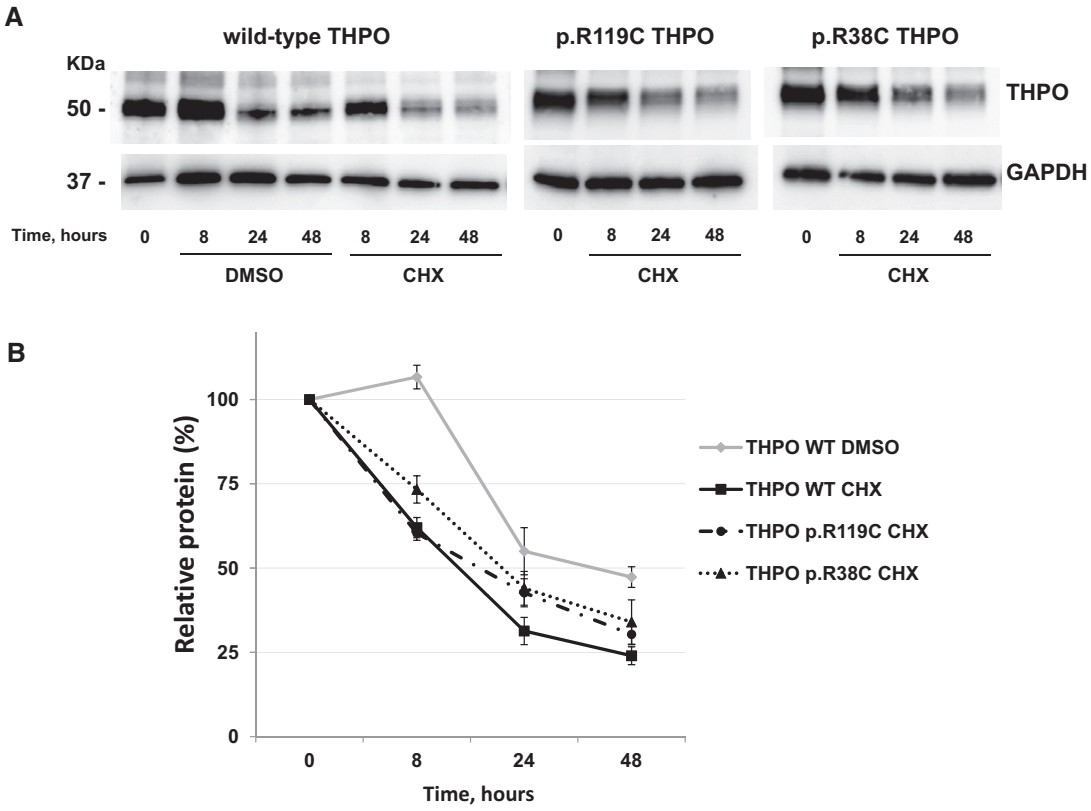

**Figure 4.   The p.R119C and p.R38C THPO proteins are stable as the wild-type THPO.**

A   Representative images of immunoblotting of HEK293T cells transfected with wild-type or mutant (p.R119C and p.R38C) *THPO*-expressing vectors. Cell lysates were prepared just before treatment with cycloheximide (CHX) (time 0) and 8, 24, and 48 h after the addition of CHX in order to block *de novo* protein synthesis (CHX chase assay). DMSO alone was used as control in the WT sample. GAPDH was used as loading control.

B   Densitometric analysis of the bands obtained in *n* = 3 independent experiments. THPO amount was measured as THPO/GAPDH ratio and expressed as the percentage of the amount measured at time 0 in each condition (Relative protein, %). Of note, the expression of wild-type THPO after CHX treatment was significantly lower compared with DMSO alone at each time point ($P < 0.05$, two-tailed Student's *t*-test), confirming that protein synthesis was efficiently blocked by CHX treatment. Data are presented as mean ± SD. The exact *P*-values are reported in Appendix Table S3.

Source data are available online for this figure.

Very recently, Seo and colleagues reported five individuals from three consanguineous families carrying another two homozygous mutations (p.R99W and p.R157*) of THPO (Seo *et al*, 2017). Like our patients, these subjects presented with hypo-megakaryocytic thrombocytopenia without or with anemia or neutropenia, which progressed to trilineage bone marrow aplasia in early childhood. Thus, these patients also had a clinical picture indistinguishable from that of CAMT caused by MPL receptor defects.

Considering that missense mutations can be regarded as variants of uncertain significance, we carried out *in vitro* studies to determine the effect of the p.R119C substitution on the THPO function. When probed at the same concentration of the wild-type protein, the p.R119C THPO was defective in sustaining the growth of a THPO-dependent cell line and in stimulating the MPL receptor. Moreover, we found that p.R119C strongly affects secretion of THPO. In fact, in cell extracts, the mutant THPO was stable and expressed at comparable levels than those of wild-type protein, but its concentration in the culture supernatants was markedly lower than that of the wild-type form. Therefore, although the amino acid substitution does not affect stability of THPO, it is likely to impair

the cellular trafficking, preventing the cytokine from being secreted. Of note, the THPO serum concentration measured in our patients was consistent with this mechanism. In fact, since the endogenous THPO is cleared from circulation by megakaryocytes and platelets, the serum or plasma THPO levels are markedly increased in all forms of bone marrow aplasia or hypoplasia, including CAMT due to MPL mutations (Dame, 2001; Ballmaier & Germeshausen, 2009; Olnes *et al*, 2012; Ballmaier *et al*, 2015). At variance, the serum THPO concentration was not increased in our patients and thus inappropriately lower than expected on the basis of their megakaryocyte–platelet mass. Like p.R119C, also p.R38C led to defective secretion of THPO, a finding that was consistent with the "normal" or low serum levels of THPO found in the other individuals carrying loss-of-function mutations of THPO (Dasouki *et al*, 2013; Seo *et al*, 2017). These observations suggest that measurement of the serum THPO level could be useful to discriminate CAMT patients with *THPO* mutations from those with *MPL* variants or other bone marrow failure syndromes.

Moreover, R119 resides in the domain that binds the MPL receptor and is highly conserved among orthologs, as well as in

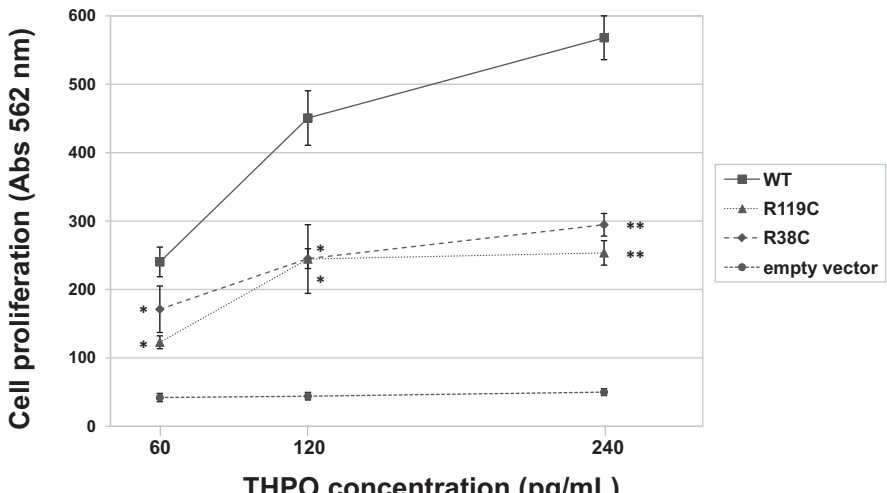

**Figure 5.  The p.R119C and p.R38C variants result in reduced functional activity of THPO protein.**

Cell proliferation measured in *n* = 3 independent MTT assays and expressed as absorbance units of UT7-TPO cells incubated with medium containing different concentrations (60, 120, or 240 pg/ml) of THPO obtained from HEK293T cells supernatants transfected with wild-type (WT) or mutant (p.R119C and p.R38C) *THPO*-expressing or empty vectors. Data are presented as means ± SD. The asterisk (*) indicates a *P*-value < 0.05, while (**) indicates a *P*-value < 0.01 with respect to WT (two-tailed Student's *t*-test). The exact *P*-values are reported in Appendix Table S4.

Source data are available online for this figure.

the human homolog EPO. Like EPO, THPO binds to the extracellular domain of MPL through two sites of different (low and high) affinity (Syed *et al*, 1998; Feese *et al*, 2004). Since residue 119 is involved in polar interactions of the low affinity site (Feese *et al*, 2004), substitution of arginine with cysteine is predicted to reduce but not abolish the binding of THPO to MPL. Consistent with this hypothesis, the functional studies demonstrated that the p.R119C THPO retains approximately 50% of the activity of the wild-type form, when measured as proliferation of THPO-dependent cells or activation of signaling kinases downstream of the MPL receptor.

THPO-mimetic drugs represent an appealing therapeutic option for the cytopenias caused by *THPO* mutations (Basciano & Bussel, 2012). Consistently, romiplostim was effective in increasing platelet count in all our patients, as well as in increasing hemoglobin concentration and neutrophil count in subjects presenting with anemia and/or neutropenia. Beside the hematological response, romiplostim induced remission of spontaneous bleeding and transfusion independence in the proband and his brother, as well as stable remission of recurrent infections in the proband. These clinical responses were maintained after a follow-up of more than 6 years in both subjects. Further observations are required to ascertain if romiplostim is able to definitively prevent recurrence of aplastic anemia in the propositus and avoid its occurrence in his two siblings. However, the prolonged trilineage response to the drug observed in the proband, despite the use of a very low dose and of a non-conventional administration schedule due to logistic reasons, allows us to hope that the drug will be effective also in this regard. Consistent with our observations, trilineage response to romiplostim was achieved by two affected individuals homozygous for the p.R99W or p.R157* mutations of *THPO* (Seo *et al*, 2017).

Distinguishing CAMT associated with *THPO* mutations from that caused by alterations of *MPL* mutations, as well as from the other forms of juvenile bone marrow failure syndrome, has relevant consequences in the clinical practice. Although HSCT is the cornerstone of treatment for these forms, transplantation is not expected to be effective in subjects with cytopenias deriving from defects in THPO. In fact, four patients with the p.R99W or p.R157* mutations underwent HSCT, which resulted in engraftment failure or persistent aplasia despite donor cell engraftment even after repeated transplantations (Seo *et al*, 2017). Three children died early after HSCT. After lack of response to HSCT despite full donor chimerism, the fourth patient was started on romiplostim obtaining a dramatic improvement of pancytopenia. Altogether, our and previous observations highlight the importance of distinguishing patients with *THPO* mutations from those with *MPL* defects or other juvenile bone marrow failure syndromes.

Of interest, a homozygous (p.R150Q; p.R177Q in the precursor protein) mutation in the *EPO* gene was recently identified in two siblings presenting with congenital hypoplastic anemia (Kim *et al*, 2017). The amino acid substitution did not affect the release of EPO, whose levels were markedly increased in patients' plasma, or the cytokine affinity for its receptor, which was only mildly reduced. Instead, it altered the ligand–receptor binding kinetics, causing impaired receptor dimerization dynamics even at maximally potent concentrations of the p.R150Q EPO. This, in turn, resulted in a significantly altered pattern of activation of the JAK2-dependent signaling effectors downstream of the EPO receptor (Kim *et al*, 2017). Thus, different mutations in the two highly homologous cytokines THPO and EPO can cause defects in the signaling pathways downstream of their receptors through very different mechanisms. Like in patients with THPO mutations, the alteration of EPO caused a disorder unresponsive to HSCT, but

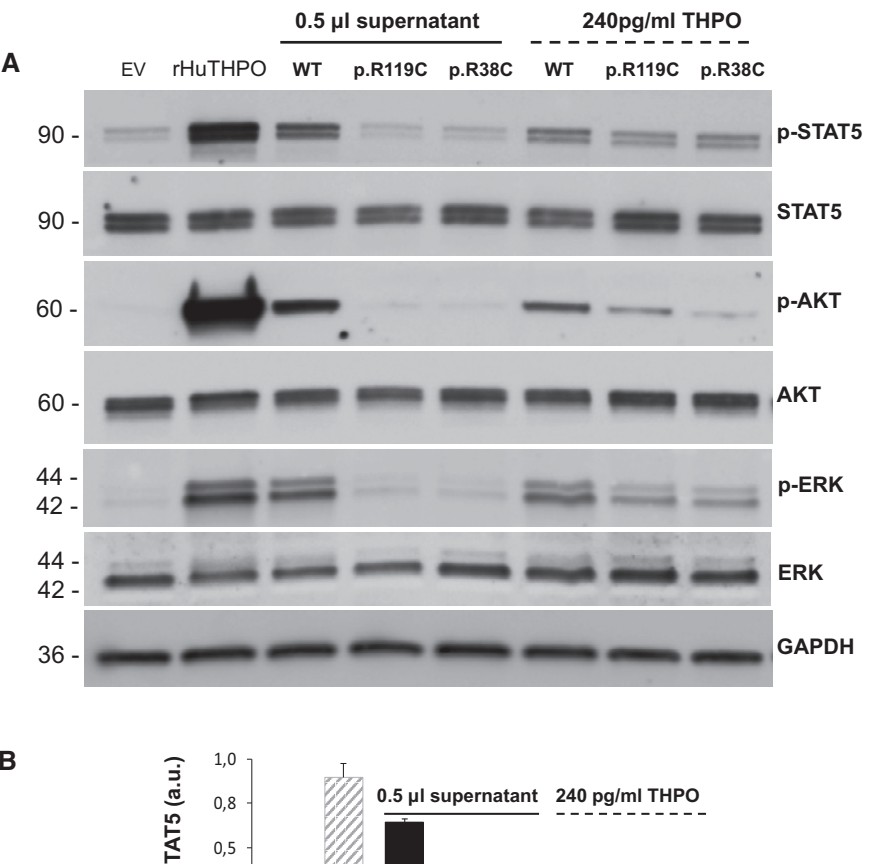

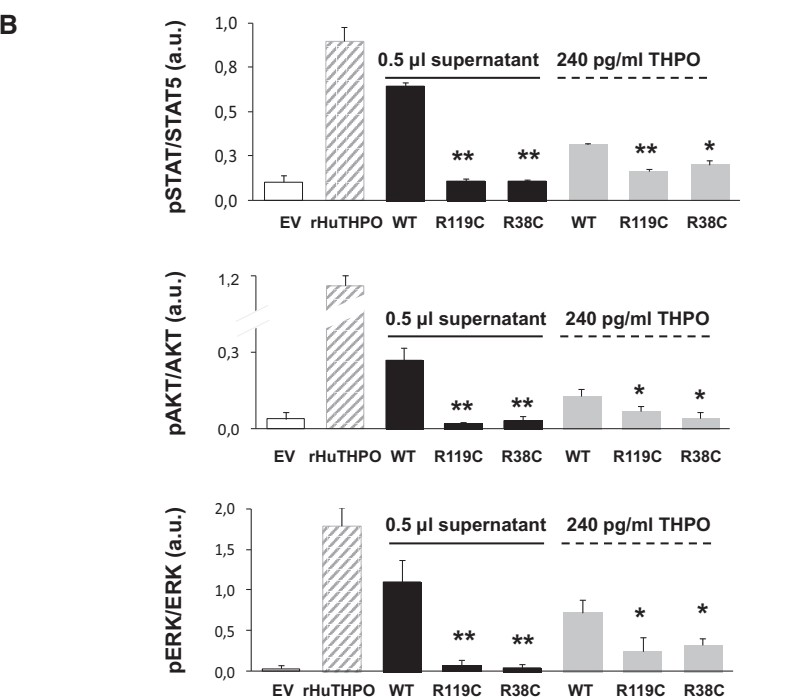

**Figure 6. Defective activation of signaling kinases downstream of the MPL receptor.**

A   Representative images of immunoblotting of UT7-TPO cells recapitulating conditions of the cell proliferation assays shown in Fig 3 (incubation with 0.5 µl of HEK293T wild-type or mutant supernatants) and Fig 5 (incubation with 240 pg/ml of THPO from HEK293T wild-type or mutant supernatants). The supernatant (0.5 µl) obtained after transfection of the empty vector (EV) was used as a negative control. Commercial recombinant human THPO (rHuTHPO) at 10 ng/ml was used as the positive control condition. The analysis investigated the phosphorylated forms of STAT5, ERK, and AKT (p-STAT5, p-ERK, and p-AKT) and the total form of the three kinases (STAT5, ERK, and AKT). GAPDH was used as loading control.

B   Densitometric analysis of the bands obtained in n = 3 independent experiments. Protein phosphorylation was expressed as the ratio between the phosphorylated and the total forms of each protein. Data are presented as means ± SD. The asterisk (*) indicates a *P*-value < 0.05, while (**) indicates a *P*-value < 0.01 with respect to WT (two-tailed Student's *t*-test). The exact *P*-values are reported in Appendix Table S5.

Source data are available online for this figure.

treatable using a substitutive therapy that replaces the activity of the defective cytokine. In fact, the proband carrying the homozygous p.R150Q EPO mutation underwent HSCT; unfortunately, he showed persistence of severe anemia despite the achievement of full donor chimerism, and died for complications stemming from the transplantation procedure. Instead, his younger sister, who shared the same genotype and clinical picture, was treated with recombinant human EPO obtaining complete remission of anemia (Kim *et al*, 2017).

In conclusion, mutations of the *THPO* gene cause a form of CAMT that can be treated with THPO-receptor agonists. The disorder should be suspected in children or young adults presenting with hypo-megakaryocytic thrombocytopenia with or without other peripheral cytopenias, including those with an apparently acquired aplastic anemia. Measurement of the serum THPO level could represent a valuable tool to identify patients with defective THPO. Recognizing these patients is essential for their correct management avoiding the use of invasive and unnecessary treatments, such as HSCT and immunosuppressive drugs.

# Materials and Methods

### Patients

Clinical investigations of the proband and his relatives were carried out from July 2009 to January 2017 at the Ain Shams University Children's Hospital, Cairo, Egypt. Romiplostim was administered at the same institution. The patients' parents signed written informed consent for the studies, for treatment with romiplostim, and for reporting their clinical data in an anonymous form. All the procedures were performed in accordance with the principles of the Declaration of Helsinki.

### Mutational screening

The *MPL* (NM_005373.2) and *THPO* (NM_000460) coding exons and the respective exon–intron boundaries were amplified by PCR. Amplification reactions were carried out in a final volume of 35 µl containing 100 ng of DNA, 17.5 µl of KAPA2G Fast Hot Start Ready Mix (Kapa Biosystems, Resnova, Cape Town, South Africa), and 1.75 µl of a 10 pmol solution of each primer (primers sequences available upon request). PCR products were sequenced using an ABI PRISM BigDye Terminator Cycle Sequencing Ready Reaction Kit and an ABI 310 Genetic Analyzer (Applied Biosystems, Foster City, CA, USA). Nucleotide A of the ATG translation initiation start site of the *THPO* gene cDNA in GenBank sequence NM_000460.3 is indicated as nucleotide +1.

### Bioinformatic analyses

The effect of the missense variations was evaluated using five pathogenicity prediction programs: PoliPhen-2 (http://genetics. bwh.harvard.edu/pph2/), Mutation Taster (http://www.mutationta ster.org/), Mutation Assessor (http://mutationassessor.org/r3/), SIFT (http://sift.jcvi.org) and, CADD (http://cadd.gs.washington.ed u). Protein structures were analyzed with the graphic program Pymol v1.3 (The PyMOL Molecular Graphics System, Schrödinger, LLC).

### Generation of *THPO* constructs

A pcDNA3.1+-C-DYK expression vector containing cDNA of human THPO (OHU10732, GenScript, Hong Kong, China) was used for mutagenesis. The c.112C>T (p.R38C) and c.335C>T (p.R119C) mutations were generated using specific primers R38C_F (5′-GTAA ACTGCTTTGTGACTCCCATGTCC-3′) and R38C_R (5′-GGACATGGG AGTCACAAAGCAGTTTAC-3′) or R119C_F (5′-CTGGACAGGTCT GTCTCCTCCTTGGGG-3′) and R119C_R (5′-CCCCAAGGAGGAGAC AGACCTGTCCAG-3′) and the QuikChange Multi Site-Directed Mutagenesis Kit (Stratagene, Agilent, San Diego, CA, USA). To confirm the wild-type and the mutant sequences, constructs were sequenced as above.

### Production of THPO-conditioned supernatants

Human embryonic kidney (HEK) 293T cells (National Institute for Cancer Research, Genoa, Italy) were maintained in Dulbecco's modified Eagle's medium with 20% fetal bovine serum (FBS) (Lonza Group Ltd, Basel, Switzerland). Aliquots of $7 \times 10^5$ cells were seeded in six-well culture plates and transfected 24 h later with the constructs harboring cDNA for wild-type THPO, p.R119C THPO, p.R38C THPO, or the empty vector, using the Lipofectamine 2000 reagent (Life Technologies, Carlsbad, CA, USA) according to the manufacturer's instructions (reagent/DNA ratio of 2.5:1). THPO-conditioned supernatants were collected 24 and 48 h after transfection, filtered, and frozen at −80°C until use. For immunoblotting, MTT or ELISA assays, media collected 24 and 48 h after transfection were pooled in a 1:1 ratio (Dasouki *et al*, 2013).

### MTT cell proliferation assay

UT7-TPO human megakaryoblastic cell line is a kind gift from Dr. Hana Raslova (Institute Gustave Roussy, Villejuif, France). UT7-TPO cells were maintained in MEM alpha medium with 10% FBS and 5 ng/ml GM-CSF (PeproTech, London, UK). After starvation for any cytokines for 16 h, aliquots of $1 \times 10^4$ cells were seeded in 96-well culture plates and incubated with 250 µl of MEM Alpha culture medium containing 0.5 or 1.0 µl of wild-type or mutant THPO-conditioned supernatants (Dasouki *et al*, 2013). The same assay was also carried out using the MEM alpha medium containing the THPO-conditioned supernatants volumes required to reach the final THPO concentrations of 60, 120, or 240 pg/ml (total volume 250 µl per well). As a negative control, the empty vector medium was used at the same volume of the medium used at the highest volume in the same experiment. Commercial recombinant human THPO (PeproTech) at 10 ng/ml, a concentration that induces maximal UT7-TPO stimulation in our experimental setting, was used in parallel as the positive control condition. After 3 days of culture, aliquots of 100 µl were transferred to a microplate and treated with 10 µl of 5 mg/ml 3-(4,5-dimethylthiazol-2-yl)-2,5-diphenyltetrazolium bromide (MTT) (Roche, Basel, Switzerland). After incubation for 6 h, 100 µl of dimethylsulfoxide was added to solubilize the MTT formazan crystals. After further overnight incubation, samples were shacked for 10 min in a titer shaker, and absorbance was measured at 562 nm by an Eppendorf BioPhotometer (Hamburg, Germany). Absorbance value of the culture medium alone was subtracted from the samples' values. Each experiment was performed in duplicate

and data presented represent the mean ± SD of three separate experiments.

## Analysis of signaling kinases downstream MPL

After starvation for any cytokines for 16 h, aliquots of $2.5 \times 10^5$ UT7-TPO cells were stimulated for 30 min at 37°C with wild-type, p.R119C, or p.R38C THPO-conditioned media and phosphorylation of ERK1/2, STAT5, and AKT was investigated by immunoblotting. To recapitulate the conditions of the proliferation assay, cells were incubated with an equal volume of wild-type or mutant supernatants (0.5 μl), or with the same concentration of wild-type or mutant THPO (240 pg/ml). The supernatant obtained after transfection of the empty vector was used at a volume of 0.5 μl as negative control. In all the cases, THPO-conditioned supernatants were diluted in MEM alpha culture buffer to reach a total volume of 250 μl per well (Dasouki *et al*, 2013). Commercial recombinant human THPO 10 ng/ml was used as the positive control condition. After washing with PBS, cells were lysed in IP buffer (10 mM Tris, 158 mM NaCl, 1% Triton X-100, 1% Na-deoxycholate, 0.1% sodium dodecyl sulfate (SDS), 5 mM EGTA, pH 7.2) with 2% protease and phosphatase inhibitor cocktail (Sigma, St. Louis, MO, USA). General procedures of immunoblotting have been previously described in details (Necchi *et al*, 2013). Membranes were probed with the following antibodies: rabbit monoclonal C11C5 anti-phospho-STAT5, diluted 1:1,000 (Cell Signaling Technology, MA, USA); rabbit monoclonal 3H7 anti-total STAT5, 1:1,000 (Cell Signaling); rabbit polyclonal anti-phospho-AKT, 1:1,000 (Cell Signaling); rabbit polyclonal anti-total AKT, 1:2,000 (Cell Signaling), rabbit polyclonal anti-phospho-ERK1/2, 1:1,000 (Cell Signaling); mouse monoclonal 137F5 anti-ERK1/2, 1:2,000 (Cell Signaling); rabbit monoclonal EPR16891 anti-GAPDH, 1:5,000 (Abcam). The appropriate HRP-conjugated secondary antibodies (Dako, Glostrup, Denmark) were used for detection. Densitometric analysis was performed by ImageJ software. Protein phosphorylation was expressed as the ratio between the phosphorylated form and the total form of each protein. Data are represented as the mean ± SD of three separate experiments.

## Immunoblotting detection of transfected THPO

Lysates prepared 24 and 48 h after transfection of HEK293T cells were pooled in a 1:1 ratio for immunoblotting (Dasouki *et al*, 2013), or analyzed separately. Membranes were probed with the following monoclonal antibodies: rabbit EPR14948 against human THPO, diluted 1:2,000 (Abcam, Cambridge, UK); mouse M2 against FLAG tag epitope, 1:1,000 (Sigma); mouse AC-15 against β-actin, 1:5,000 (Sigma). The appropriate HRP-conjugated secondary antibodies (Dako, Glostrup, Denmark) were used for detection. Densitometric analysis was performed by ImageJ software. Immunoblotting was performed on HEK293T cells obtained by the same three separate transfection experiments performed for the collection of THPO-conditioned supernatants for ELISA and MTT assays. Data represent the mean ± SD of all experiments.

## Cycloheximide chase assay

Cycloheximide chase assay for investigation of stability of THPO proteins was performed as previously reported (Zhang *et al*, 2016;

### The paper explained

#### Problem
Congenital amegakaryocytic thrombocytopenia is an inherited disease characterized by severe low platelet count since birth, which is due to absence of megakaryocytes (progenitor cells of platelets), and progression to aplastic anemia during childhood. CAMT is always fatal unless children are treated with HSCT, the only treatment available so far. Mutations of *MPL*, the gene encoding for the receptor of thrombopoietin, have previously been associated with CAMT but not in all affected individuals, suggesting that other genes have yet to be identified.

#### Results
In a family with three children affected with CAMT, we excluded the presence of mutations in the *MPL* gene. Instead, we found that a novel mutation in the *THPO* gene, which encodes for thrombopoietin, was the cause of the disease. The mutation induces low thrombopoietin production and defective binding of thrombopoietin to its receptor MPL, suggesting that both quantitative and qualitative defects of thrombopoietin are responsible for the disease. The three children were treated with romiplostim, a drug that mimics the effects of thrombopoietin, and thus is expected to compensate the above defects. All of them obtained a significant improvement of blood cell counts, remission of bleeding and infections, and independence from transfusions, which was maintained after an up to 6.5-year observation.

#### Impact
Patients with CAMT caused by *THPO* mutation can be effectively treated with romiplostim. Considering that thrombopoietin is mainly produced by the liver and other cells extrinsic to hematopoietic cells, these patients are not expected to benefit from hematopoietic stem cell transplantation. Therefore, distinguishing patients with *THPO* mutations from those with *MPL* mutations is of fundamental importance not only to give the appropriate treatment, but also to avoid the use of unnecessary and risky procedures.

Marconi *et al*, 2017). Briefly, cycloheximide 100 μg/ml diluted in DMSO was added to HEK293T cells 24 h after transfection with wild-type, p.R119C, or p.R38C THPO, in order to block *de novo* protein synthesis. Control conditions were carried out by adding the same amount of DMSO alone to the HEK293T transfected with wild-type THPO. Cells were collected and lysed just before the addition of cycloheximide or DMSO (time 0) and 8, 24, and 48 h after the addition of cycloheximide or DMSO. Proteins were analyzed at each time point by immunoblotting as reported above. GAPDH was used as loading control. THPO amount was measured as THPO/GAPDH ratio and expressed as the percentage of the amount measured at time 0 in each condition. Data represent the mean ± SD of three separate experiments.

## ELISA measurement of THPO concentration

THPO concentration in patients' plasma or in the THPO-conditioned supernatants was assessed by the Quantikine ELISA–Human Thrombopoietin Immunoassay Kit (R&D system, Minneapolis, MN, USA) according to the manufacturer's instructions. Each sample was assayed in duplicate. Optical density was read at 450 nm with wavelength correction at 540 nm on a microplate reader model 680 (Bio-Rad). ELISA assay was performed on the conditioned media of

HEK293T derived from three different transfection experiments. Data represent the mean ± SD of all experiments.

Expanded View for this article is available online.

## Acknowledgements

The authors thank Dr. Hana Raslova, Institute Gustave Roussy, Villejuif, France, for providing the UT7-TPO cell line. This study was supported by the ERA-Net for Research Program on Rare Diseases (EUPLANE), Cariplo Foundation (2012-0529), and Italian Ministry of Health (RF-2010-2309222).

## Author contributions

APe, CLB, and AS designed research, interpreted data, and wrote the manuscript; IR, HA, and MS investigated patients and interpreted clinical data; VB, DDR, SB, TG, FM, CA, and APa performed research, collected and interpreted data; all the authors critically revised the manuscript and accepted the final version.

## Conflict of interest

The authors declare that they have no conflict of interest.

## For more information

OMIM database:

CAMT: http://omim.org/entry/604498?search=CAMT&highlight=camt

MPL: http://omim.org/entry/159530

THPO: http://omim.org/entry/600044?search=THPO&highlight=thpo

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
