## [Review Process File · EMBO Molecular Medicine]

Thrombopoietin mutation in congenital amegakaryocytic thrombocytopenia treatable with romiplostim

Alessandro Pecci, Iman Ragab, Valeria Bozzi, Daniela De Rocco, Serena Barozzi, Tania Giangregorio, Heba Ali, Federica Melazzini, Mohamed Sallam, Caterina Alfano, Annalisa Pastore, Carlo L. Balduini, and Anna Savoia

Corresponding author: Anna Savoia, Institute for Maternal and Child Health - IRCCS Burlo Garofolo

Review timeline:

Submission date:	16 June 2017
Editorial Decision:	20 July 2017
Revision received:	15 September 2017
Editorial Decision:	23 October 2017
Revision received:	27 October 2017
Accepted:	02 November 2017

Transaction Report:

Editors: Roberto Buccione and Céline Carret

1st Editorial Decision

20 July 2017

Thank you for the submission of your manuscript to EMBO Molecular Medicine. We have now heard back from the three Reviewers whom we asked to evaluate your manuscript.

As you will see the reviewers are very positive and suggest a number of improvements to increase the impact and quality of presentation.

I will not delve into any detail, as their evaluations are quite straightforward and clear. I would just mention that reviewer 3 would like you to provide further data in support of your contention that there is an impairment of THPO secretion. Also, reviewers 2 and 3 would both like more context and discussion on the use of romiplostim and also suggest you incorporate a discussion of the Kim et al, 2017 paper (PMID: 28283061).

In conclusion, while publication of the manuscript cannot be considered at this stage, we would be pleased to consider a revised submission, with the understanding that the Reviewers' points must be addressed including further experimentation as necessary. Eventual acceptance of the manuscript will entail a second round of review.

I look forward to seeing a revised form of your manuscript as soon as possible.

***** Reviewer's comments *****

Referee #1 (Comments on Novelty/Model System):

This is a well written and interesting manuscript. It is only the second family in whom a recessive defect in thrombopoietin is described. The long-term follow-up of 6 years of treatment are impressive, the report is clinically highly relevant for these very rare cases, and the laboratory experiments convincingly show the functional consequences of the mutation. My comments are only minor. Due to the rarity of such mutations, the manuscript will be very frequently cited in the literature on thrombopoiesis.

The medical impact is high, because this is only the 2nd family described with the genetic mutation leading to THPO deficiency. It is important to diagnose those patients, as the standard therapy for a plastic anemia/thrombocytopenia would be bone marrow transplantation which is inefficient in these patients

The model is appropriate, as the transfected cell lines show impaired release of THPO, which is fully consistent with the low levels of the protein measured in patient blood

Referee #1 (Remarks):

Pecci et al. describe the second family with inherited deficiency of thrombopoietin (THPO) causing congenital amegakaryocytic thrombocytopenia. The recessive disorder in the family is caused by a homozygous mutation in p. R119C. This leads to moderately reduced function of THPO, normal THPO production in transfected cell lines, but impaired release of the protein into the circulation. The clinical consequences are the same as in other forms of CAMT, with the major difference that THPO levels are either reduced or normal but not excessively increased like in patients with a mutation in the receptor for THPO. Treatment with the thrombopoietin receptor agonist romiplostim resulted in improvement of hematopoietic cell counts. The manuscript is interesting and very relevant for the differential diagnoses of causes of CAMT and the choice of treatment.

Minor comments:

page 2, abstract

line 9: the terminus "production" seems not to be correct in the view of the experiments with HEK cells, in which the production was normal but the release was impaired.

Line 11: the abstract states that infections had been reduced in the affected children, while this information is missing in the results section. Please add information on the infection frequencies in the description of patients in the section reporting on treatment response.

Page 5, para 3, line 5 and 7: the cell culture was performed in 96 well culture plates. Did you really incubate the cell culture with 0.5 μ L only? Was the mutant media diluted in any culture buffer?

Page 7, para 3: please give information on infection frequencies

Page 8, para 1, line 2: The abbreviation gr/dl is unusual, g/dL is mostly used, please double check with the journal style.

Page 8, para 2, line 3: reports = report

page 9, para 3, line 2: again the question of the 0.5 or 1.0 μ L volume

page 13, last para: the para is a copy of the first para

figures:

figure 3: legend Y axis Proliferation = proliferation;

figure 5 A and B: supernatant = supernatant; 0,5 = 0.5 (also accounts for the legend). Please use larger font size for molecular weight standard

supplementary figure 1: please consider to use multiple symbols to show presence of thrombocytopenia, infection, anemia and neutropenia, e.g. by showing 4 little black or white squares within the large square symbol

Referee #2 (Remarks):

In this paper, the authors report in Egyptian family with three children affected by thrombocytopenia, associated variably with anemia and pancytopenia, that have homozygous mutations altering the amino acids sequence of the cytokine thrombopoietin (THPO). The specific mutation is in a conserved domain of the protein, which is found in low amounts in the supernatant

of cells engineered to express the mutant THPO. In addition, the serum levels of THPO, while at normal levels, are functionally low considering the degree of thrombocytopenia. It is not known whether this mutation leads to any abnormalities in the binding of the mutant THPO to mpl. Most importantly, a clinical response was evident with the use of the THPO-mimetic romiplostim. While this response is highly clinically relevant, the intermittent nature of the medical follow-up appears to be responsible for the highly variable blood counts shown in Figure 1. However, taken together, the data support the conclusions and the discussion points raised by the authors.

Minor suggestions:

1. I would suggest moving the data from supplemental figure 3 into figure 3 of the paper. It will be useful to show the normal protein levels of the mutant THPO proteins in contrast to the levels in the supernatants of the HEK293 cells.
2. It would be useful to discuss the recent very elegant paper (Kim, et al. Cell 2017) showing that a mutation in erythropoietin is responsible for altered binding to its receptor and to altered signalling downstream of that receptor in erythroid cells. In addition, this paper describes the tragic death of the proband following unnecessary bone marrow transplantation prior to the discovery of the underlying mutation.

Referee #3 (Remarks):

In this interesting manuscript from Pecci and colleagues, a novel mutation in THPO causing congenital amegakaryocytic thrombocytopenia (CAMT) in multiple members of a single family is described. This mutation reduces the expression of THPO and thus appears to cause disease. Importantly, the authors show that romiplostim therapy can treat this rare disorder. While this manuscript is interesting and important, several issues must be addressed prior to publication:

1. The presentation of the manuscript is confusing. For example, there are experiments involving a p.R38C mutant THPO that come before the reference describing this mutation. The introduction should be revised to summarize known mutations more clearly. It is also unclear what prompted the authors to initiate therapy with romiplostim. This needs to more clearly be explained. Why not escalate romiplostim dosing?
2. The authors suggest that there is an impairment of protein secretion. Can the authors be sure that there is not a problem with protein stability and not secretion? Can further analysis of this lesion be performed? This would help the published findings tremendously.
3. The authors should reference a recent paper discussing similar mutations in other CAMT patients (Seo et al., Blood, 2017).
4. The authors should also compare and contrast this mutation with other mutations in hematopoietic cytokines. For example, a recent paper described an EPO mutation with altered downstream signaling properties in patients with congenital hypoplastic anemia (Kim et al., Cell, 2017). These other similar mutations should be referenced and discussed more fully to put these findings in context. The EPO case is particularly interesting, given the response to recombinant EPO, similar to what is observed here.

1st Revision - authors' response

15 September 2017

Referee #1 (Comments on Novelty/Model System):

This is a well written and interesting manuscript. It is only the second family in whom a recessive defect in thrombopoietin is described. The long-term follow-up of 6 years of treatment are impressive, the report is clinically highly relevant for these very rare cases, and the laboratory experiments convincingly show the functional consequences of the mutation. My comments are only minor. Due to the rarity of such mutations, the manuscript will be very frequently cited in the literature on thrombopoiesis.

The medical impact is high, because this is only the 2nd family described with the genetic mutation leading to THPO deficiency. It is important to diagnose those patients, as the standard therapy for a

plastic anemia/thrombocytopenia would be bone marrow transplantation which is inefficient in these patients. The model is appropriate, as the transfected cell lines show impaired release of THPO, which is fully consistent with the low levels of the protein measured in patient blood. Pecci et al. describe the second family with inherited deficiency of thrombopoietin (THPO) causing congenital amegakaryocytic thrombocytopenia. The recessive disorder in the family is caused by a homozygous mutation in p. R119C. This leads to moderately reduced function of THPO, normal THPO production in transfected cell lines, but impaired release of the protein into the circulation. The clinical consequences are the same as in other forms of CAMT, with the major difference that THPO levels are either reduced or normal but not excessively increased like in patients with a mutation in the receptor for THPO. Treatment with the thrombopoietin receptor agonist romiplostim resulted in improvement of hematopoietic cell counts. The manuscript is interesting and very relevant for the differential diagnoses of causes of CAMT and the choice of treatment.

R: We thank the reviewer for the encouraging and detailed comments on our work.

Minor comments:

- page 2, abstract

line 9: the terminus "production" seems not to be correct in the view of the experiments with HEK cells, in which the production was normal but the release was impaired.

R: The referee is right. The abstract has been modified according to the reviewer's suggestion and to the author guidelines (not more than 175 words). The word "production" does not appear any longer.

- Line 11: the abstract states that infections had been reduced in the affected children, while this information is missing in the results section. Please add information on the infection frequencies in the description of patients in the section reporting on treatment response.

- Page 7, para 3: please give information on infection frequencies

R: We thank the reviewer for his/her suggestion. Before romiplostim administration, the patient had episodes of febrile neutropenia, or fever and otitis media, requiring hospitalization every 1-2 months. Conversely, no further infectious episodes were recorded during romiplostim administration. The missing information was included in the revised version of the manuscript (Results: page 4, second paragraph, lines 2-3; page 5, first two lines).

- Page 5, para 3, line 5 and 7: the cell culture was performed in 96 well culture plates. Did you really incubate the cell culture with 0.5 μ L only? Was the mutant media diluted in any culture buffer?

- Page 9, para 3, line 2: again the question of the 0.5 or 1.0 μ L volume

R: The cells were incubated in 96-well plates with 250 μ L of culture buffer containing 0.5 or 1.0 μ L of THPO-conditioned supernatants from HEK293T cells cultures (wild-type, mutant, or empty vector). The culture buffer was MEM Alpha medium. The methods of the experiments were the same as those previously reported in the study by Dasouki et al. on the effects of the p.R38C mutation (Dasouki *et al.*, 2013). We used the same protocol to be able to reproduce their results and compare the effects of the p.R119C variant with those of the p.R38C.

We understand that the previous version of the paper might have been a bit confusing on this point. We have thus modified the manuscript with more accurate explanations (Results: page 6, second paragraph of "The p.R119C affects both secretion of THPO and THPO/MPL signalling", lines 2-4; Patients and Methods: page 13, in "MTT cell proliferation assay", lines 3-7; page 14, in "Analysis of signaling kinases downstream MPL", lines 6-8; legend of Figure 3).

- Page 8, para 1, line 2: The abbreviation gr/dl is unusual, g/dL is mostly used, please double check with the journal style.

Page 8, para 2, line 3: reports = report

page 13, last para: the para is a copy of the first para

R: We apologize for the typo and thank the reviewer for the helpful indications. The manuscript was modified accordingly.

figures:

figure 3: legend Y axis Proliferation = proliferation; figure 5 A and B: surnatant = supernatant; 0,5 = 0.5 (also accounts for the legend). Please use larger font size for molecular weight standard
 supplementary figure 1: please consider to use multiple symbols to show presence of thrombocytopenia, infection, anemia and neutropenia, e.g. by showing 4 little black or white squares within the large square symbol

R: The figures were modified according to the reviewer's suggestions.

Referee #2 (Remarks):

In this paper, the authors report in Egyptian family with three children affected by thrombocytopenia, associated variably with anemia and pancytopenia, that have homozygous mutations altering the amino acids sequence of the cytokine thrombopoietin (THPO). The specific mutation is in a conserved domain of the protein, which is found in low amounts in the supernatant of cells engineered to express the mutant THPO. In addition, the serum levels of THPO, while at normal levels, are functionally low considering the degree of thrombocytopenia. It is not known whether this mutation leads to any abnormalities in the binding of the mutant THPO to mpl. Most importantly, a clinical response was evident with the use of the THPO-mimetic romiplostim. While this response is highly clinically relevant, the intermittent nature of the medical follow-up appears to be responsible for the highly variable blood counts shown in Figure 1. However, taken together, the data support the conclusions and the discussion points raised by the authors.

R: We thank the reviewer for the favorable and specific comments.

Minor suggestions:

1. I would suggest moving the data from supplemental figure 3 into figure 3 of the paper. It will be useful to show the normal protein levels of the mutant THPO proteins in contrast to the levels in the supernatants of the HEK293 cells.

R: Figure 3 shows the proliferation of the UT7-TPO cells induced by the THPO-conditioned supernatants of the HEK293T cells. The reduction observed for p.R119C and p.R38C is not only due to the low level of the mutant THPO in the supernatant but also to defective activity in stimulating the signaling downstream of MPL, as shown in Figures 5 and 6. For this reason, we think it is more appropriate to move supplementary Figure 3 into Figure 2 of the paper. The new Figure 2 recapitulates all the immunoblotting experiments to investigate the THPO expression in HEK293T cells and now shows that the expression level of the different forms of THPO is maintained for 24 and 48 hours after transfection, the time points at which the supernatants were collected for cell proliferation assays. Accordingly, in the revised version of the manuscript Figure 2 and its relative legend were modified.

2. It would be useful to discuss the recent very elegant paper (Kim, et al. Cell 2017) showing that a mutation in erythropoietin is responsible for altered binding to its receptor and to altered signaling downstream of that receptor in erythroid cells. In addition, this paper describes the tragic death of the proband following an unnecessary bone marrow transplantation prior to the discovery of the underlying mutation.

R: We thank the reviewer for his/her suggestion. We have included a paragraph commenting on the paper by Kim and colleagues in the Discussion section (page 10, third paragraph).

Referee #3 (Remarks):

In this interesting manuscript from Pecci and colleagues, a novel mutation in THPO causing congenital amegakaryocytic thrombocytopenia (CAMT) in multiple members of a single family is described. This mutation reduces the expression of THPO and thus appears to cause disease. Importantly, the authors show that romiplostim therapy can treat this rare disorder. While this manuscript is interesting and important, several issues must be addressed prior to publication:

1. The presentation of the manuscript is confusing. For example, there are experiments involving a p.R38C mutant THPO that come before the reference describing this mutation. The introduction should be revised to summarize known mutations more clearly.

R: We thank the reviewer for the constructive comments, which have undoubtedly helped us to have a significant improvement of the manuscript. The introduction was modified according to the reviewer's suggestion. Reference to the p.R38C mutation and the other two *THPO* mutations very recently identified by Seo *et al* have been now included (Introduction: page 3, third paragraph).

It is also unclear what prompted the authors to initiate therapy with romiplostim. This needs to more clearly be explained.

R: The proband was started on romiplostim due to lack of a donor for hematopoietic stem cell transplantation and of any other available therapeutic options - as an empirical attempt to increase platelet count in the child who had severe, life-threatening hemorrhages and required very frequent platelet transfusions. Of note, the proband initially received the diagnosis of idiopathic aplastic anemia and, before empirical romiplostim administration, he was treated with cyclosporine A for 4 months, without any response, and then with oxymetholone, again without any improvement. The two siblings of the proband were started on romiplostim in view of the efficacy of the treatment in the proband.

Following the reviewer's suggestion, the manuscript has been revised to clarify these aspects (Results: page 4, third paragraph, lines 1-5).

- Why not escalate romiplostim dosing?

R: Romiplostim was administered to the three children using a non-conventional schedule and low doses (4 µg/Kg/monthly) because of difficulties of the family in accessing healthcare facilities in their country (Egypt), as reported in the manuscript (page 4, third paragraph, lines 6-7). For the same reason, it was not possible to escalate the romiplostim dose. Nevertheless, all the three children maintained sustained hematological and clinical responses.

2. The authors suggest that there is an impairment of protein secretion. Can the authors be sure that there is not a problem with protein stability and not secretion? Can further analysis of this lesion be performed? This would help the published findings tremendously.

R: We thank the reviewer for this very appropriate suggestion. To investigate whether the p.R119C (and p.R38C) substitution could affect stability of the THPO protein, we studied stability of the wild type and mutant forms by the standard cycloheximide chase assay. After blocking the *de novo* protein synthesis in HEK293T cells by treatment with cycloheximide, the kinetics of degradation of the wild type and mutant THPO proteins were assessed over 48 hours. The assay showed that the two mutant proteins have similar stability to that of the wild type form. These new data have been included in the revised version of the manuscript (Results: page 7, second paragraph; Patients and Methods: page 15, "Cycloheximide chase assay"; new Figure 4 with relative legend).

3. The authors should reference a recent paper discussing similar mutations in other CAMT patients (Seo *et al.*, Blood, 2017).

R: We thank the reviewer for the suggestion that gave us the opportunity to comment on this interesting paper. Accordingly, the revised manuscript has been modified to include the main findings reported by Seo and colleagues (Introduction: page 3, third paragraph, lines 4-5; Discussion: page 8, third paragraph, lines 7-12; Discussion: page 9, first paragraph, lines 14-15; Discussion: page 10, first paragraph, lines 4-5 and second paragraph, lines 4-10).

4. The authors should also compare and contrast this mutation with other mutations in hematopoietic cytokines. For example, a recent paper described an EPO mutation with altered downstream signaling properties in patients with congenital hypoplastic anemia (Kim *et al.*, Cell, 2017). These other similar mutations should be referenced and discussed more fully to put these findings in context. The EPO case is particularly interesting, given the response to recombinant EPO, similar to what is observed here.

R: We thank the reviewer for his/her suggestion. We have included a paragraph commenting on the paper by Kim and colleagues in the Discussion section (page 10, third paragraph).

2nd Editorial Decision

23 October 2017

Thank you for the submission of your revised manuscript to EMBO Molecular Medicine. We have now received the enclosed reports from the referees that were asked to re-assess it. As you will see the reviewers are now supportive and I am pleased to inform you that we will be able to accept your manuscript pending editorial amendments.

***** Reviewer's comments *****

Referee #2 (Remarks for Author):

The authors have responded effectively to all of the suggestions/critiques of the reviewers with the addition of data, changes to the text, including reorganization of the presentation of data and a discussion of other papers that helps to put this paper in a wider context.

Referee #3 (Comments on Novelty/Model System for Author):

This paper is much improved and I have no additional comments

Referee #3 (Remarks for Author):

Very nice revision.

Corresponding Author Name: Anna Savoia
 Journal Submitted to: EMBO Molecular Medicine
 Manuscript Number: EMM-2017-08168-V2